# Body Temperature—Indoor Condition Monitor and Activity Recognition by MEMS Accelerometer Based on IoT-Alert System for People in Quarantine Due to COVID-19

**DOI:** 10.3390/s21072313

**Published:** 2021-03-26

**Authors:** Minh Long Hoang, Marco Carratù, Vincenzo Paciello, Antonio Pietrosanto

**Affiliations:** Department of Industrial Engineering, University of Salerno, 84084 Fisciano, SA, Italy; mcarratu@unisa.it (M.C.); vpaciello@unisa.it (V.P.); apietrosanto@unisa.it (A.P.)

**Keywords:** COVID-19, body temperature, IoT, accelerometer, thermometer, wearable device, indoor condition, activity recognition

## Abstract

Coronavirus disease 19 (COVID-19) is a virus that spreads through contact with the respiratory droplets of infected persons, so quarantine is mandatory to break the infection chain. This paper proposes a wearable device with the Internet of Things (IoT) integration for real-time monitoring of body temperature the indoor condition via an alert system to the person in quarantine. The alert is transferred when the body thermal exceeds the allowed threshold temperature. Moreover, an algorithm Repetition Spikes Counter (RSC) based on an accelerometer is employed in the role of human activity recognition to realize whether the quarantined person is doing physical exercise or not, for auto-adjustment of threshold temperature. The real-time warning and stored data analysis support the family members/doctors in following and updating the quarantined people’s body temperature behavior in the tele-distance. The experiment includes an M5stickC wearable device, a Microelectromechanical system (MEMS) accelerometer, an infrared thermometer, and a digital temperature sensor equipped with the user’s wrist. The indoor temperature and humidity are measured to restrict the virus spread and supervise the room condition of the person in quarantine. The information is transferred to the cloud via Wi-Fi with Message Queue Telemetry Transport (MQTT) broker. The Bluetooth is integrated as an option for the data transfer from the self-isolated person to the electronic device of a family member in the case of Wi-Fi failed connection. The tested result was obtained from a student in quarantine for 14 days. The designed system successfully monitored the body temperature, exercise activity, and indoor condition of the quarantined person that handy during the Covid-19 pandemic.

## 1. Introduction

Since 2020, the term “quarantine” has become familiar due to the global pandemic of coronavirus disease 19 (COVID-19) [1]. Self-isolation is a way to prevent infection between people via the respiratory. After traveling from another country/region, people need to stay at home and separate themselves from the family members for at least 14 days [2]. High body temperature (fever) is the most common symptom of COVID-19 [3]. Therefore, the people in quarantine must have their temperature checked frequently. However, many older people and children are not able to check their temperature accurately.

Moreover, it is challenging to monitor body temperature during sleep or accidentally forgetting to check. Therefore, IoT systems’ help has played a critical role in the healthcare system for real-time monitor [4]. On the other hand, a wearable device is a useful tool for thermal body measurement. As discussed in [5], the authors propose that wearable sensors can detect illnesses by continuous fever monitoring. The headset module’s work Integrated to mask system was presented in [6] for the integrated health monitoring via Wi-Fi. Another work uses a headset attached to the microphone/speaker input (via 3.5 mm jack) and detects breathing problems, respiration rate, and cough. Although these designs contribute to combat COVID 19 significantly, the concern about activity with the quarantined people’s significant temperature variation still needs to be considered deeply. Moreover, Wi-Fi is not always available due to unexpected disconnection, service maintenance, or poor internet connection. Therefore, this paper develops a more complete wearable system that considers various temperature monitors for the quarantined people and indoor conditions to restrain the virus spread at home.

According to the advice of the World Health Organization (WHO), physical activity is a valuable method to help quarantined people remain calm and continue to protect their health during this time [7]. However, body temperature increases during physical exercises [8]. It is necessary to realize this activity and update the temperature threshold (T_TH_) for alert temperature because the high-temperature warning works depending on the comparison between these temperature parameters. Usually, the human activity recognition (HAR) algorithm is used for the prediction of the movement of a person based on sensor data [9] by using Artificial Intelligence such as Machine Learning [10] and Deep Learning [11,12]. It requires much training data and complexity. Thus, this paper proposes a simple algorithm, Repetition Spikes Counter (RSC), which counts the number of spikes per a specified period (Ts) to detect whether the person is doing the physical exercise or not. The T_TH_ will be automatically updated from this activity recognition to be suitable with the current activity condition. The MEMS accelerometer [13,14,15] inside the wearable device provides the acceleration value. The spikes are calculated based on the acceleration variation as the input of the RSC model.

The average human body temperature stays around 36.5 °C to 37 °C, regardless of the external temperature or weather [16]. Each person’s body is different, but generally, when muscles are primed and ready to go, the body’s optimum core temperature falls within a specific range; it is right around 37.5 degrees Celsius [17]. Regarding room conditions, the ambient temperature and humidity are critical factors for the place where has the quarantined people. To restrain the spread of Covid-19, the indoor temperature (Tindoor) should be maintained between 20 °C and 24 °C [18], while relative humidity (RH) should be maintained between 20% and 60% [19,20,21]. In this project, a wearable device based on M5stickC [22] is integrated containing a small-size infrared thermometer MLX90614 [23] for body temperature and ENV HAT [24] for the indoor temperature and relative humidity (RD).

The IoT solutions connect the devices by transferring and receiving the data with each other [25,26,27], where wide range of sensors [28] are used to measure various physical phenomena The two most popular wireless network protocols—Wi-Fi and Bluetooth—are implemented into the proposed system, so the users are able to select the desired mode freely. All the data can be observed and stored easily by doctors and family members. If Wi-Fi has a connection issue, Bluetooth still sends the data to the relatives at home without physical contact.

The alert message is transferred to the user via the liquid-crystal display (LCD), Cloud via MQQT broker, or Bluetooth via Serial Bluetooth terminal. The user can know their body temperature every time and appropriately adjust their exercise activity to avoid high temperature.

Overall, the proposed system operation is based on the digital thermometer, with the accelerometer under IoT protocol. The main objectives of the work are to make a coherent combination between both contact and non-contact thermometers for updating the temperature routine during quarantine with high accuracy. The essential feature of this device is to provide warnings if body temperature is greater than a threshold value which is renovated, following the user activity, and detected by a new algorithm RSC. Another strong point of this design is the flexibility of data communication with Bluetooth, Wi-Fi, and LCD. In this project, the incorporation of all the single parts from the algorithm’s sensors assembly to data transmission helps the wearable device collect and store thermal data in real-time with activity recognition practically. By this way, the thermal body info of the quarantined person is always up to date for the user and all the relatives/doctors with the advantages of less complexity and high effectiveness.

## 2. Material and Methods

The IoT currently has primary healthcare usage for controlling and managing dire situations during the coronavirus disease (COVID-19) [29]. The IoT node sends the sensor data to the smartphone or PC via Bluetooth connection. Another communication protocol is to send the data stream to the server via 4G/5G or Wi-Fi. The cloud server receives all the transferred information and stores them for a specified time. At this point, the doctor or relatives can observe the thermal parameter, activity of the self-isolated person in real-time, or analyze the historical data to judge the health situation.

MQTT is an Outcome and Assessment Information Set (OASIS) standard messaging protocol for the Internet of Things (IoT), which has been used widely in the medical monitoring system as discussed in [30]. In [31], the authors have addressed the design MQTT protocol for temperature control in a warehouse using ESP8266 and Wi-Fi. The obtained data are shared to the Baidu intelligent cloud server using MQTT. In the proposed design, the MQTT protocol is implemented into ESP32 [32] via open-source Arduino Software (IDE) [33] for messaging transfer between wearable device and cloud.

Wearable electronic devices have been used in the early detection of asymptomatic and pre-symptomatic cases of COVID-19 [34]. During the quarantine or self-treatment period, a person/patient can be fully monitored with wearable assistive technology [35]. The combination between the wearable device and IoT is the best solution for health monitoring of the people in quarantine as it would take a long time before the vaccines are permitted for public use popularly over the world.

To keep a balanced body state during quarantine, physical exercise is an indispensable activity that also needs monitoring technology to guarantee this progress’s safety. The MEMS accelerometer is an effective tool in activity recognition by allowing the signal variation to reflect humans’ corresponding action [36]. In this paper, the approach focuses on distinguishing between regular activity and physical exercise, so the applied algorithm is supposed to be simple but effective. The normal body temperature threshold is automatically adjusted to follow the involved recognition as the T_body_ during and after training usually is higher than usual. The exercise recognition helps the observer to understand the variation of the thermal body of the isolated person.

The thermometer is the indispensable sensor because fever is the most apparent symptom of COVID-19 infection. The contact thermometer is equipped with the user to follow the T_body_, archiving high accuracy directly, but it requires a long holding time. The infrared thermometer [37] consumes a short time to display the T_body,_ but it is less precise than the contact one. This project approaches both methods: the contact-digital thermometer is mounted with the wearable device for long-time usage. The infrared thermometer is connected to the device used when the user wants to free the hand from the wearable device.

On the other hand, the sensor for room condition is also necessary to assure the suitable living condition for the quarantined person and restrain the virus’s spread to others. The correlation between thermal body and ambient was addressed in [38] with the integration of an infrared thermometer and a capacitive humidity sensor for remote ambient and people who pass this designed platform. In our solution, the ambient sensor is mounted on the wearable device directly, which continuously checks the ambient temperature and RH surrounding the user. All the data are shown to the user via the device screen.

By leveraging the known correlations between indoor conditions and body state, this approach could substantially improve the monitor capability for the quarantine cases without requiring access to large training datasets and without requiring access to large training datasets. In the warning situation, the alert is displayed to the user via the LCD screen and sent to the observers via IoT communication since the T_body_ exceeds the allowed threshold. The significant advantage of the proposed system is low-cost, small-size, and multi-function, as shown in Figure 1. 

### 2.1. Room Condition Sensor

The T_indoor_ and RH are two crucial factors to prevent the spread of COVID-19. As discussed early, the advised T_indoor_ should be maintained between 20 °C and 24 °C, while RH should be maintained between 20% and 60% [18]. The ambient sensor measures the surrounding environment’s temperature and humidity and converts the input data into the corresponding electrical signal to monitor the indoor changes. The quarantined person can adjust the room condition by modifying the air conditioner, heater, or air humidifier based on the sensor data. In the winter, the temperature is low and drier, especially in Europe, so the family of the quarantined person should take into account the indoor conditions. The ambient temperature and humidity surrounding the quarantined person are updated each 5 s using ambient sensor, as shown in Figure 2.

### 2.2. Body Temperature Sensor

There are two main categories of thermometers: contact and non-contact. Contact thermometers measure temperature by the heat transfer phenomenon, which requires physical contact with the measured object for the object temperature measurement. Meanwhile, an infrared sensor measures the body temperature by reading the level of infrared emissions.

#### 2.2.1. Infrared Thermometer

According to the Stefan–Boltzman law [39], any object is above absolute zero (0° K) that emits (non-human-eye-visible) light in the infrared spectrum. An infrared thermometer infers temperature from a portion of the thermal radiation emitted by the object being measured. This sensor type is contactless. It includes a lens to focus the infrared (IR) energy on to a detector known as a thermopile, which converts the heat energy to an electrical signal. Voltage output is produced in proportion to the incident infrared energy, which is used to determine the temperature, then displays the accurate temperature [40].

This sensor supports the system to monitor the thermal body of people in quarantine. In this project, the MLX90614 [23] is selected, made up of an IR thermopile detector and signal conditioning shown in Figure 3. The voltage regulator regulates 5 V as the constant voltage source.

The MLX90614 operation is controlled by an internal state machine, which calculates the object temperatures, then postprocesses the signal through the pulse width modulation (PWM).The output of the IR sensors is amplified by a low-noise chopper amplifier (OPA) with programmable gain, converted by a high-resolution Analog–Digital Converter (ADC) to a single bitstream and fed to a digital signal processing for further processing.At the next stage, the signal passes through the low pass filter to remove the high-frequency signal and refresh rate. Finally, the measurement result is available in the internal random-access memory.

#### 2.2.2. Digital Temperature Sensor

Unlike the infrared thermometer, the digital temperature sensor requires physical contact with the user to sense the electrical signal. Then, it converts convert the analog voltage to a digital word. This sensor type is suitable for long-time contact like the wearable device as the quarantined people can use it frequently. In this application, the temperature sensor MCP9808 [41] is chosen with the best benefit of low-cost and stable measurement with less noise.

The MCP9808 uses a bandgap temperature sensor circuit to output analog voltage proportional to absolute temperature. An internal analog to digital converter is used to convert the analog voltage to a digital world, as shown in Figure 4. 

In the designed system, both thermometer types are utilized, as illustrated in Figure 5. The main thermometer is MCP9808, connected to the wearable device, and it measures the body temperature directly via physical contact. Due to the limit of battery time, the infrared sensor is used during the charging time or when the user wants to remove the wearable device. The MLX90614 is also embedded with the device, so the quarantined person can quickly take this infrared thermometer to measure the body temperature by pressing the button of the device. The measured temperature data will be sent to the cloud or the shared device automatically based on the programming implementation. Practically, the contact thermometer achieves superior accuracy due to long-term measurement, but the infrared sensor has the advantages of quick measurement with the no-contact requirement. Moreover, a low-pass (LP) filter [42,43] is necessary for the contact thermometer to remove the noise due to vibration. Here, a first-order infinite impulse response filter that uses a weighting coefficient to decrease the noise exponentially, close to zero.
(1)$$\;T_{out}\;=\;\beta \;T_{i}+\left(1-\;\beta \right)\;T_{i-1},$$
where
*T_out_* is the output of the LP filter.*T_i_* and *T_i_*
_− 1_ are the temperature input from the thermometer.The coefficient β represents the degree of weighting decrease, a constant smoothing, β ∈ [0, 1). If β = 1, the output is equal to the input without any filtered signal. A smaller value of β makes the filtering effect stronger, but it increases the filter’s time constant. In this work, the selected β is 0.5 to guarantee sufficient filtering capability and still maintain a good dynamic response.

#### 2.2.3. Thermometer Calibration

To achieve good measurements, the digital thermometers were calibrated by using the glass thermometer, which is traditionally used for body temperature checking, as shown in Figure 6. This regular thermometer has trustable indication which requires significant time of body contact to provide the result.

There are two main steps for the temperature calibration:At the first process, the digital sensor is equipped on the wrist and the glass thermometer is placed inside the body simultaneously. After 10 min, the glass thermometer reaches the proper value and becomes the temperature reference (Tref).At this stage, the digital thermometer already receives the heat of user skin and acquires the stable measurement. Since this moment, the calibration procedure for digital thermometer is carried out by averaging N samples from raw data (Traw). This average result (Tavg) contains any bias such as skin heat. Now, the difference between Tref and Tavg is calculated as bias to be removed from Traw.

By this way, the calibration successfully removed the bias from the digital measurements as expressed in following equations:(2)$$\mathrm{Tavg}\;=\;\frac{\sum_{i=1}^{N}Ti}{N}$$
Tbias = Tref − Tavg(3)
Tcal = Traw + Tbias(4)
where
Tavg is the average result from *N* sample number of the digital thermometer. In this calibration, *N* = 50 for average calculation.*Ti* is the digital temperature data.Tbias is the temperature difference between wrist and the inside body part such as armpit.Tcal is the calibrated temperature measurement which is output of the sensor device.

### 2.3. RSC Based on MEM Accelerometer

To realize the human activity, it is necessary to acquire and analyze the data from acceleration. In this case, the Z-axis (Zacc) acceleration is used to calculate the absolute difference between its two-consecutive sample (ΔZacc) with Zn, Zn − 1 are the previous and current Zacc, for activity recognition.

Figure 7 shows the behavior of Zacc and |ΔZacc| during three activity conditions:No motion (sleep, sit at one place, etc.).Normal activity (stands up, normal behavior, etc.).Doing exercise.

After the first 30 s of start-up, the Zacc signal acquisition was carried out for 30 s in real time. A low-pass filter was applied to remove the noise and limit the accelerometer’s vulnerability, so the signal only reacts to the apparent motion of the body.

Under the no-motion condition, the Zacc maintains constant value without any spikes, so the |ΔZacc| is always almost zero. Regular activity can cause some sudden variations, but it is easy to realize that the signal behavior is random with small |ΔZacc| at low frequency. On the other hand, if the user does physical exercises, the Zacc frequently varies with high repetitive spikes due to considerable motions that cause |ΔZacc| changes repetitively with significant value.

To distinguish between normal activity and training, the RSC algorithm counts the number of spikes per a specified period (Ts) (30 s in this case) to decide whether the person is doing the physical exercise or not. Moreover, the body’s temperature typically returns to normal within 20 min after the exercise has been completed [44], so a timer is set to count this period.

There are three adjusted thresholds, which can be chosen by the user:Number of spikes per Ts (counter)The ΔZacc value which is considered as spikes (ΔZ_TH_).Threshold of spike number per Ts, which is considered as doing exercise (count_TH_).Temperature threshold for body (T_TH_)

For the proposed system, Ts = 30 s; ΔZ_TH_ = 90 mg; count_TH_ = 24. These values are suitable for exercise recognition, which can be modified depending on the user. However, if the user trains extremely soft exercise, it will not be enough to increase the T_body_. In this case, the user desires to consider this activity as exercise, so ΔZ_TH_ and count_TH_ can be decreased manually. Smaller ΔZ_TH_ realizes softer motion and vice versa, count_TH_ relates to motion speed. ΔZ_TH_ = 90 mg is suitable threshold to realize the strong action as training motion as shown in Figure 7.

At this point, the RSC algorithm practically updates the T_TH,_ following the state of physical activity as shown in Figure 8. If the quarantined person receives the warning message, the training should be ended. This feature prevents the user from risky point by overtraining since he/she is in sensitive period.

### 2.4. Output System

#### 2.4.1. A liquid-Crystal Display (LCD)

A liquid-crystal display (LCD) is embedded in the wearable device to display the data and the message for the user, like in Figure 9. The current temperature always shows on the screen, and the alert message works as the following principle:If T_body_ < T_TH_ → Normal conditionIf T_body_ ≥ T_TH_ → Warning message

To guarantee alert reliability, the alert only activates when T_body_ ≥ T_TH_ continuously for a specified time which is set for 12 min in this case.

The device continuously shows the current information to the user: T_body_ and Temperature Condition. The quarantined person can update their situation every time effectively.

#### 2.4.2. IoT Solution

The data are delivered by IoT solution for healthcare client. The information will be sent to the cloud, observed by anyone who is granted permission, like in Figure 10. The Wi-Fi-and-Bluetooth combo chip ESP32 is chosen tool for wireless communication. All necessary data are monitored with tele-distance:Alert message is demonstrated under Boolean: 1 means the warning. The red light will be shown on the display of the watching system. 0 indicates the typical condition of body thermal.Body temperature: Tbody (°C)Room Condition (Tindoor °C and RH %)Exercise Recognition (Boolean): 1—Doing Exercise; 0—Normal Activity. This info helps the observers to follow and understand the activity and body thermal clearly.

### 2.5. System Overview

Figure 11 summarizes the total operating procedures of the designed system. The MEMS accelerometer has the role of acquiring acceleration which is used for the activity recognition by RSC model. The TTH is adjusted automatically then compared with the Tbody for a body temperature check from the recognition stage. The ambient sensor collects the indoor statistics. The alert message is sent in the case of high temperature occurs continuously for a specified time. All the necessary data are sent to the LCD screen for the user and other shared devices via Bluetooth or Wi-Fi stored in the cloud.

### 2.6. Experimental Setup

All the sensors are coupled with the wearable device M5stickC like Figure 12, which is equipped with the user. The involved sensors in the experiment are as follows:The M5stickC is a development platform, which includes flash memory 4 MB and a Lipo-battery of 95 mA-3.7 V, which the IDE can program. This board contains the MEMS 3-axis accelerometer MPU6886. In this application, the acceleration range is ±2 g and the acquired frequency is 20 Hz after the low-pass filter.DHT12 [45] measures the ambient temperature and humidity sensor, connected to the Arduino Uno board via I^2^C (Inter-Integrated Circuit) as the serial communication protocol. DHT12 is embedded in the ENV HAT to be compatible with M5stickC.The thermal body sensor MLX90614 is connected to M5stickC via I^2^C serial communication.The LCD screen displays the essential information: body temperature, thermal condition, and user activity.The data are sent to the cloud via MQTT broker [46]. This method is advantageous for saving battery, network band, and enhancing the real-time experience. In this work, the data are hosted by the Ubidots [47], which is an IoT data analytics and visualization company. The transferred data can be observed by anyone who is allowed by the user.Although the accuracy can increase after a long time of contact with the body, the measurement uncertainty of the contact thermometer is unavoidable. Therefore, once the T_body_ exceeds T_TH_ for 12 min continuously, the infrared thermometer should be activated to guarantee the data precision. If both sensors indicate the high temperature, the fever symptom is truly considerable in this situation.

## 3. Results

### 3.1. Result and Analysis

The device was tested by a volunteer student who came from another country and stayed in quarantine for 2 weeks. The data were monitored and recorded for analysis. The experiment was carried out for 14 days. Each day the proposed device was used three times, each time prolongs the total duration by about 45 min due to battery charging and other activities which must be undertaken to remove the wearable appliance such as taking a bath, etc.

The experimental evaluation is divided into three main parts:Overview system interactionThe 1st part examines all the transferred data to the cloud in detail of the last quarantine day.The 2nd part summarizes all the most important during the whole quarantined period. From these statistics, the temperature comportment, as well as the alert situation, are investigated.

### 3.2. Wireless System Interaction

Data on the web server via Wi-Fi

Figure 13 shows the Overview of the observed data on the Ubidots web. At this time, the infrared thermometer was used to measure the body temperature at noon. The room temperature is normal, but the RH is a little bit low, with respect to the ideal condition (20% to 60%). The quarantined person can lower the heater or place water containers on elevated surfaces to increase humidity in a room. The information indicated that no physical exercise is occurring and no alert since the T_body_ is normal.

Data on the shared Smartphone via Serial Bluetooth Terminal

Figure 14 demonstrates the communication between the wearable device and smartphone via Bluetooth. The smartphone receives the concerned information in real-time, including the exercise status, the allowed T_body_, T_body_, as well as T_indoor_ and RH.

### 3.3. Data Analysis of Last Quarantine Day

On the last day of the quarantine, the user reported the device was worn as the following schedule. The quarantined person also tried his best to use the device for the almost maximum battery capacity each time with a collaborative schedule. The battery can last about 45 min.

In the morning: from 9:00 a.m. to 9:45 a.m.In the afternoon, the user wears while doing physical exercise for approximately 35 min.At night, the user holds from 9:00 p.m.to 9:45 p.m.

Figure 15 illustrates the data of the T_body_ after three repeats of measurements. Although the digital thermometer has variation, the T_body_ is still always in the allowed range without any noticeable points. The temperature after exercising was slightly increased, compared with other times. At night, the user temperature returns to routine temperature, close to the morning temperature. Due to the multiple activities, the T_body_ has variability. The standard deviation (Std) is about 0.2 °C, which shows the variation of the measurement, with respect with average temperature. However, the average temperature is always under the T_TH_, so there is no alert message reported in Table 1. 

The user also measures the temperature several times by the infrared thermometer while taking off the device. The device was programmed, so the user just needs to hold the infrared sensor on the forehead and then press the device button to take T_body_. All the data are transferred to the cloud automatically. The body temperature is normal without any high temperature point in the measurement of Table 2. 

Exercise analysis:

In this part, the exercising time is analyzed as Figure 16. The red line shows whether the quarantined person is doing physical exercise by using Boolean (1: Yes; 0: No) as described early. In this period, the T_body_ has a larger variation and a higher mean value than regular activity. After the exercise, the T_body_ still increases and gradually reduces later. With the information of exercising status, the family members/doctor can understand this temperature growth.

Indoor condition

The data of indoor conditions were also collected to visualize in Figure 17. The RH is a little bit lower than expected humidity, especially in the afternoon due to European winter’s dry atmosphere. However, the quarantined person already tried to adjust the room conditioner for better RH at night. The T_indoor_ is quite good, which is suitable for living conditions, as recorded in Table 3.

### 3.4. Data Overview of Quarantine Period

The wearable device data were obtained to calculate the average T_body_ of the last time use per day. It is vital to judge the thermal body to guarantee everything is under control at the end of the day. Figure 18 and Table 4 show the average temperature of the last measured time every day (usually at night). There are seven values of T_body_ of the last measurement each 14 days. All of the values are located in the normal range from 36.3 °C to 36.9 °C which does not exceed the T_TH_ (37 °C).

These indicated data demonstrate that the person in quarantine has normal thermal body without fever. The monitoring system successfully tracks the quarantined person’s temperature condition, which is helpful under the scope of the Corona pandemic.

After using the device for quarantine period, the user reported that the device has provided the adequate information for the user and the relatives. The family members were able to follow the temperature body as well as the indoor conditioning of the quarantined person, so they can rest assured about the user’s health during the pandemic. On the other hand, it also requires good courage to wear this device every day in this concerned period because it is not always convenient to wear the devices.

## 4. Conclusions

The designed system successfully monitors the user temperature and indoor condition. The numerous measurements were acknowledged from the self-isolated person due to COVID-19. Furthermore, IoT enabled connecting sensor data to the internet with the MQTT protocol via Bluetooth and Wi-Fi. With wireless communication, the information always reaches other shared devices, which is extremely essential since the virus can infect others via the respiratory. The project’s main aim is to use the low-cost and small-size device equipped with the body conveniently for supervising the health of the body whether fever occurs or not. The adoption of the proposed device also suggests a suitable way to combine both contact and non-contact thermometer during the quarantine period. The paper also describes the RSC algorithm based on acceleration for exercise recognition, followed by the auto-adjustment of threshold temperature. The alert is always ready in case the thermal body of the user exceeds this threshold. Besides, the room temperature and humidity are monitored to prevent the virus from spreading and guaranteeing the appropriate living conditions during the quarantine.

In the future, the designed device will be tested with more people to learn more about the pros and cons of the system. The obtained measurements from various people can support the project in developing the device in terms of temperature precision and the signal communication, which are really useful not only in the case of Covid quarantine, but also medical care in general.

## Figures and Tables

**Figure 1 sensors-21-02313-f001:**
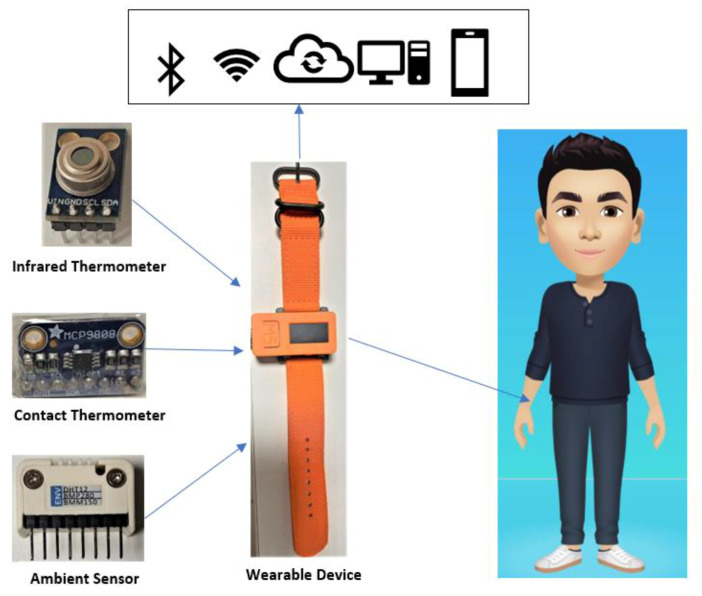
Monitoring system during quarantine.

**Figure 2 sensors-21-02313-f002:**
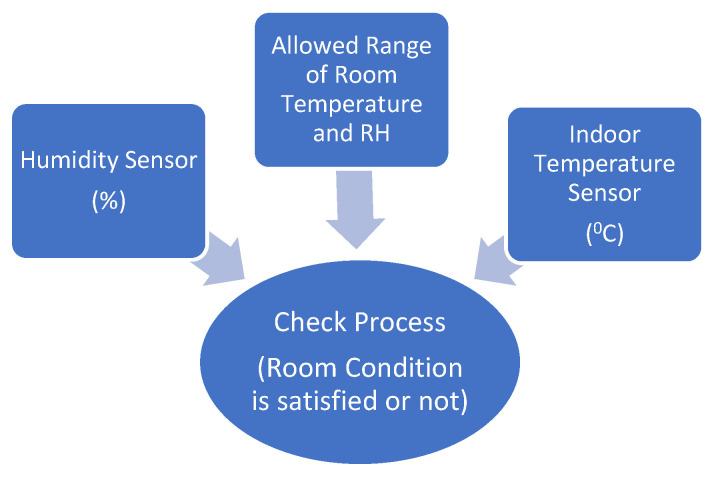
Ambient sensor in operating system.

**Figure 3 sensors-21-02313-f003:**
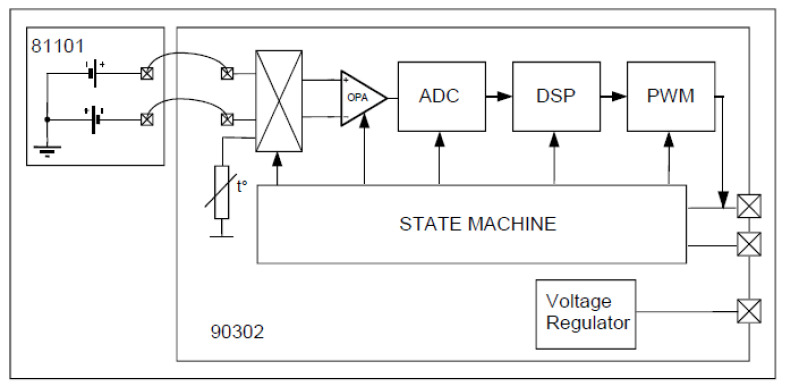
Block diagram of the IR sensor [23].

**Figure 4 sensors-21-02313-f004:**
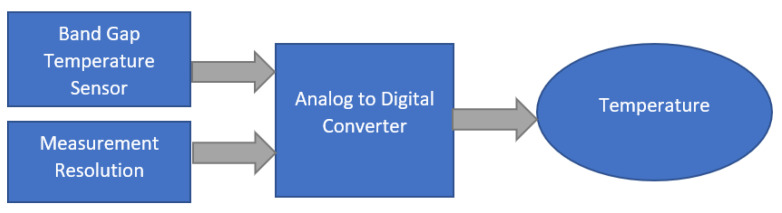
Digital thermometer diagram.

**Figure 5 sensors-21-02313-f005:**
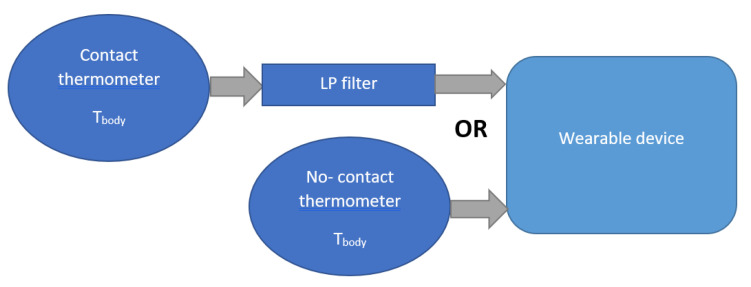
Connection between the thermometer and wearable device.

**Figure 6 sensors-21-02313-f006:**
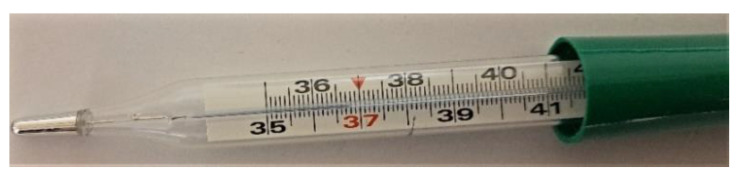
Glass thermometer measurement.

**Figure 7 sensors-21-02313-f007:**
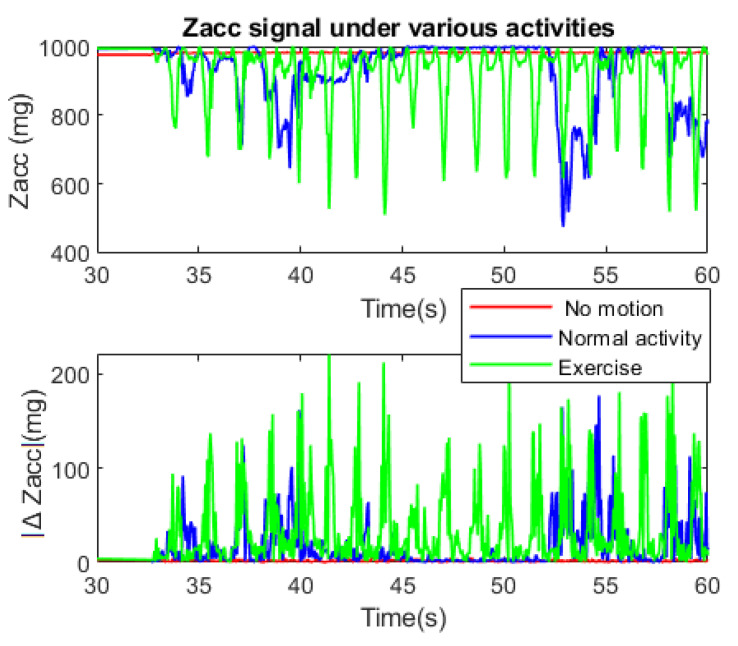
Zacc signal under various activity.

**Figure 8 sensors-21-02313-f008:**
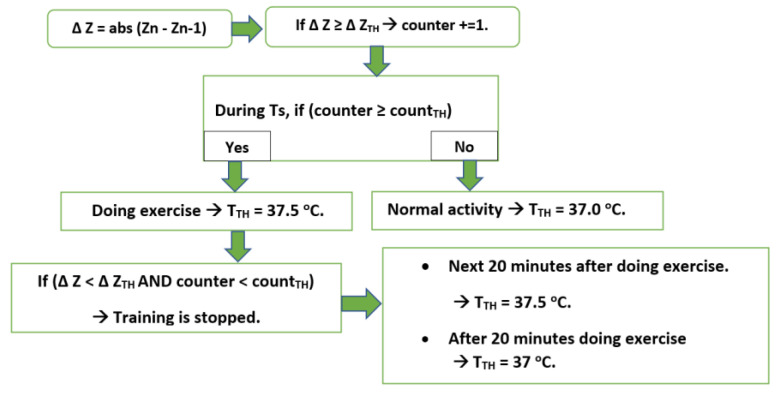
Diagram of threshold update based on activity.

**Figure 9 sensors-21-02313-f009:**
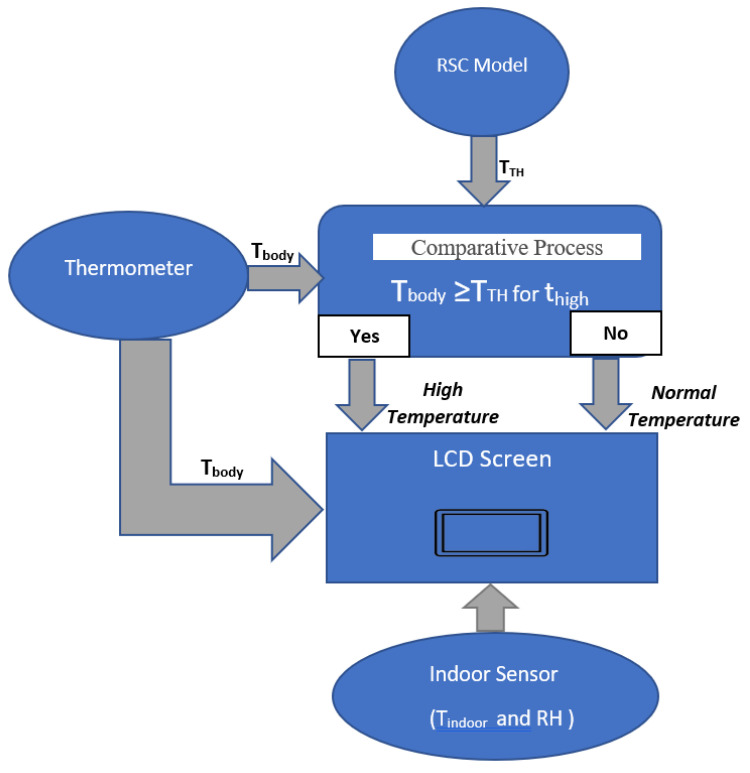
Displayed info for user.

**Figure 10 sensors-21-02313-f010:**
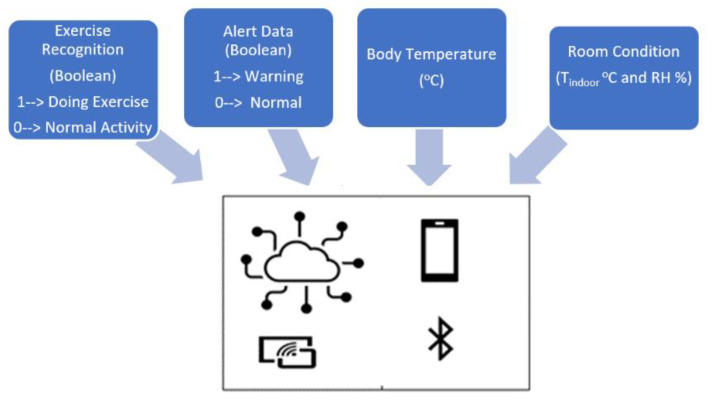
Shared data among devices via wireless communication.

**Figure 11 sensors-21-02313-f011:**
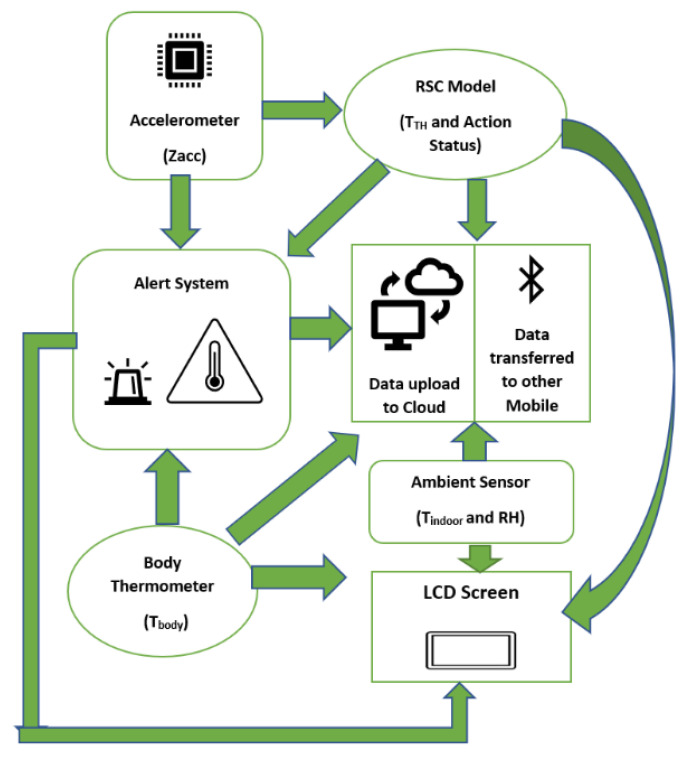
Overview of designed system.

**Figure 12 sensors-21-02313-f012:**
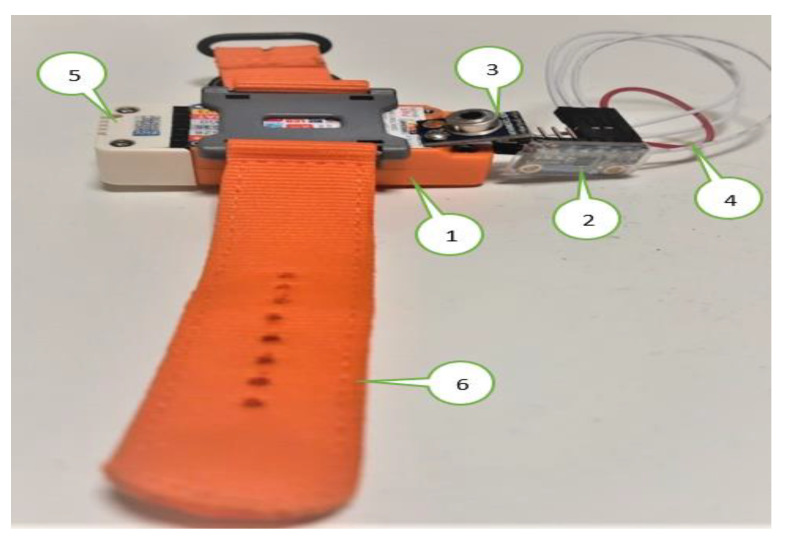
Wearable device and connected sensors.; 1. ESP32 M5StickC Development Board with LCD Display; 2. Digital thermometer: MCP9808; 3. Infrared thermometer: MLX90614; 4. Serial cable; 5. Indoor sensor ENV Hat; 6. Strap.

**Figure 13 sensors-21-02313-f013:**
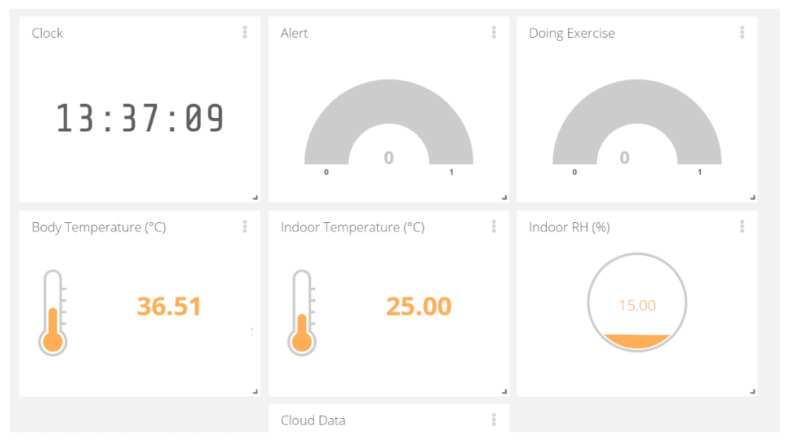
Monitored data on shared web server.

**Figure 14 sensors-21-02313-f014:**
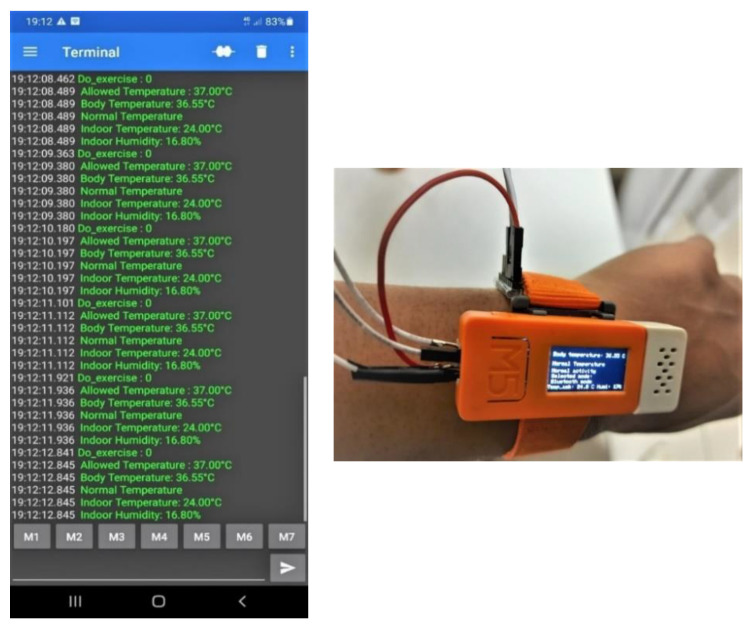
Device on wrist and transferred data on smartphone via Bluetooth.

**Figure 15 sensors-21-02313-f015:**
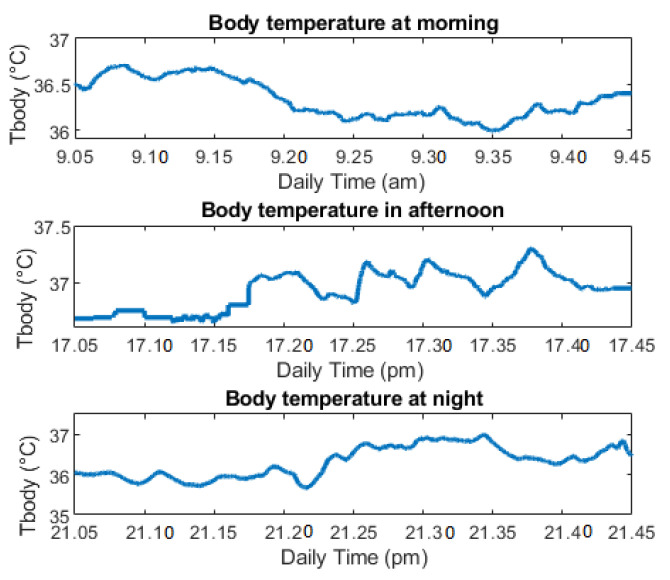
Body temperature of last quarantine day.

**Figure 16 sensors-21-02313-f016:**
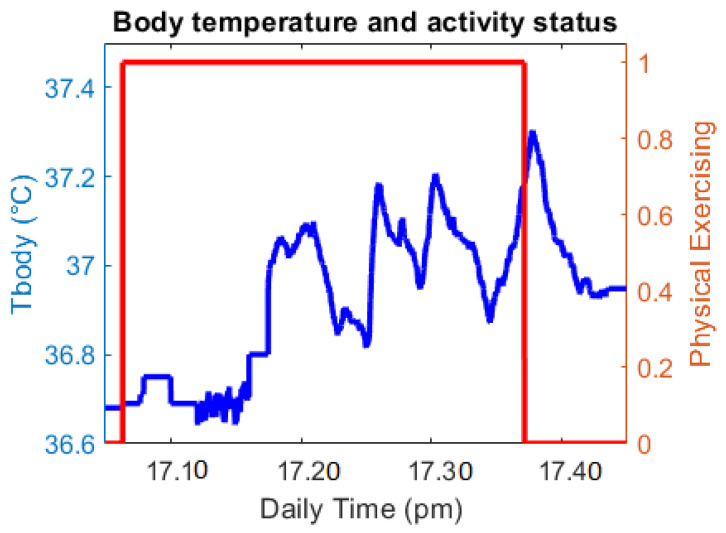
Body temperature and exercise status.

**Figure 17 sensors-21-02313-f017:**
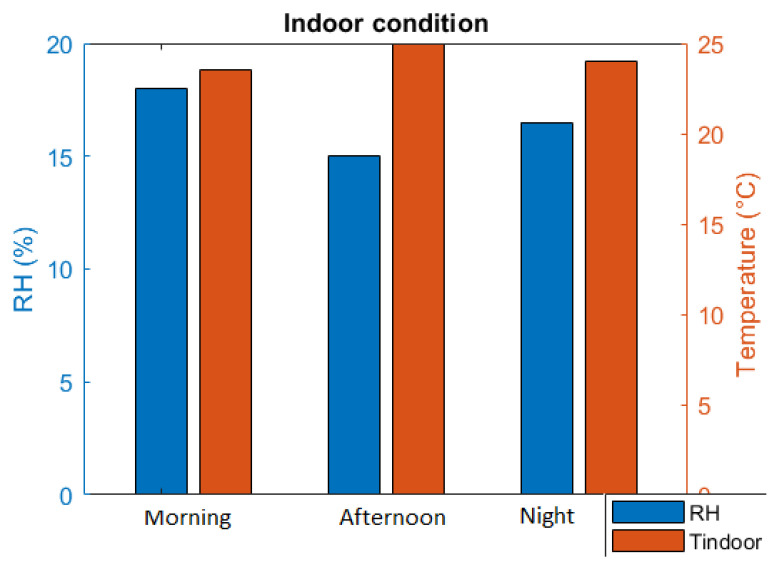
Room condition.

**Figure 18 sensors-21-02313-f018:**
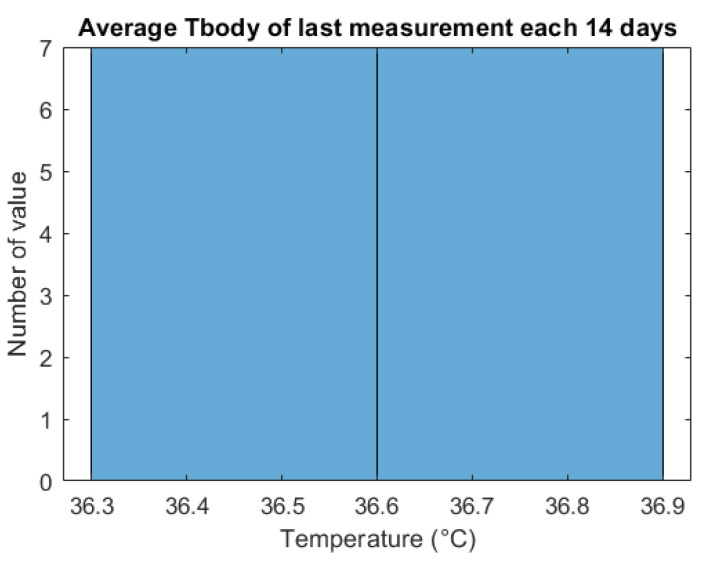
Body temperature range of quarantined person.

**Table 1 sensors-21-02313-t001:** Body temperature monitor by contact thermometer.

Parameter of T_body_	Morning (Normal Activity)	Afternoon (Exercise Activity)	Night (Before Sleep)
Minimum (°C)	35.99	36.64	35.72
Maximum (°C)	36.70	36.71	36.90
Average (°C)	36.34	36.90	36.33
Std (°C)	0.16	0.21	0.19
T_TH_ (°C)	37	37.5	37
Alert	No	No	No

**Table 2 sensors-21-02313-t002:** Body temperature by infrared thermometer.

Daily Time	T_body_ (°C)	Alert
11:18 a.m.	36.51	0
2:25 p.m.	36.49	0
4:07 p.m.	36.70	0
7:58 p.m.	36.68	0

**Table 3 sensors-21-02313-t003:** Reported data of room condition.

Average Value	Morning	Afternoon	Night
RH (%)	18	15	16.5
T_indoor_ (°C)	23.5	25	24

**Table 4 sensors-21-02313-t004:** Averaged body temperature in the last measurement each day.

Index Day	Average T_body_ (°C)	Alert
1	36.82	0
2	36.70	0
3	36.70	0
4	36.45	0
5	36.70	0
6	36.58	0
7	36.33	0
8	36.70	0
9	36.78	0
10	36.33	0
11	36.69	0
12	36.45	0
13	36.58	0
14	36.33	0

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
