# Peer review of "Body Temperature—Indoor Condition Monitor and Activity Recognition by MEMS Accelerometer Based on IoT-Alert System for People in Quarantine Due to COVID-19"

_sensors, 2021, doi:10.3390/s21072313_

Round 1
Reviewer 1 Report
This paper provides a structured presentation of an IoT wearable for use in Quarantine. Overall, the paper is well-written and organized. However, the scientific significance of the work has not been addressed. More specific:
- it is not obvious what the innovation of the work is. Authors could clarify the motivation and innovation of the work both in abstract and n introduction.
- In addition, the motivation behind the development of a wearable. It seems that ready-made wearables could provide the same kind of information (IR, PH, accelerometer, T). Authors could better explain the innovation of their approach .
- Is section 2 needed to be so long? Maybe authors could consider presenting the info in a tabular form giving more space to the algorithm description. It would be beneficial if application flowcharts could be used for the description of the algorithm (e.g., line 246).
- Section 3 seems more as data overview rather that data analysis. Perhaps authors could rearrange the section in order to support the motivation of the work.
Author Response
Thank you for your comments which are really valuable for us to improve our manuscript. Please see the attachment for our responses.

Reviewer 2 Report
The authors have created a wearable device for monitoring body temperature for individuals under quarantine. The device monitors body temperature and provides warnings if it is greater than a threshold value. Additionally, the device monitors activity and reset the threshold temperature value during and for a set duration following activity period. The device transmits measured values via Bluetooth and WiFi to the cloud as well as a mobile app.
The authors have done a nice job of incorporating different sensors and communication modules to perform this task but there are certain shortcomings in the current version of the manuscript.
Some sentences need to be revised for the sake of clarity. Also, the writing/choice of words in some places are awkward. I suggest careful examination of the writing with some revisions. For example,
Line#206-207: Moreover, a low-pass (LP) filter [40,41] is necessary for the contact thermometer to remove the vibration from the vibration due to various activity.
Line# 218-219: The Zacc signal acquisition was carried out in real-time, starting from the second 30 and prolonging for another 30s.
Line#245: Generally, the idea is to track the proper threshold for the thermal body.
Line#247-249
The authors have selected to use wrist temperature as the surrogate for body temperature. The human body does not have a uniform temperature such that the temperature at different location is affected by internal and external factors, such as local heat generation and exposure to cold or warm environment. These factors affect local temperature to different extent depending on the multiple factors including body part and distance from the skin surface. For skin surface the measured temperature would be greatly affected with the environmental conditions specially when the skin is not covered with clothing that could provide thermal insulation. Please see Nigel Taylor’s 2014 publication on measurement of body temperature. Furthermore, both core body and skin temperature follow a circadian rhythm. Please see Krauchi and Wirz-Justice 1994. The diurnal pattern of skin temperature is different between glabrous and nonglabrous region where it follows the diurnal pattern of core body temperature in nonglabrous regions and is reversed in the glabrous region. The skin temperature on the ventral surface of wrist, at the proximity of radial artery would probably follow the circadian pattern of palmar skin (glabrous skin) temperature whereas skin temperature on the dorsal surface of wrist should follow the circadian pattern of change in core temperature. All these should be considered when body temperature is reported for the purpose of determining existence of fever. For circadian rhythm of wrist temperature please see Sarabia and Mendiola 2008.
The major points that need to be considered and discussed are
- If the wrist temperature is a correct representation of core body temperature
- If the threshold temperature could be considered constant in the absence of physical exercise.
- Is 37C an accurate representation of wrist T under normal conditions?
- How does the magnitude of skin temperature depend on the skin coverage for example wearing long sleeves that cover the skin vs a short sleeve leaving skin exposed to ambient
Moreover, the accuracy of skin temperature measurement is dependent on the quality of contact between the temperature source and skin. Please discuss the measures taken to make sure the skin temperature is measured accurately.
Please provide accuracy and precision associated with each mode of temperature measurement sensor (IR and contact).
Authors indicate that patient data will be transmitted via Bluetooth and WiFi. Is the data encrypted? what type of security measured is used as the data include patient information.
Could the measured ambient temperature have been affected by body temperature given the proximity of ambient temperature sensor to patient’s body?
The battery life seems to be very short. This might be due to the continuous data transmission. One way to improve battery life could be by reducing the frequency of data transmission. This could be via burst transmission or transmitting an average value for each measured parameter over a chosen time span.
Line#164-165 Authors state “This spectrum is directly proportional to its temperature.” Strictly speaking, total energy is related to the fourth power of T, so not a linear relationship. Of course, the self-radiation of the infrared thermometer should also be considered in the calculation of temperature. The power would be different (not a fourth power) if the IR thermometer does not cover all range of wavelength.
Why is only movement in Z axis is considered? Why not include x axis or y axis? Why not all three?
Authors indicate that a low pass filter was used for Zacc data. Please provide the filter characteristics.
What is thigh in line# 257?
It is not clear why both Bluetooth and WiFi are used. Why not transmit data to a mobile device via Bluetooth and then upload to cloud from the mobile device? This could reduce the complexity of design and power usage.
The authors report data from a human subject experiment but there is no indication of approval for a human subject study from an institutional review board (IRB). Authors might want to remove this data if the experiment actually did not have an IRB approval or indicate why an IRB approval was not required. In the absence of an IRB approved protocol or IRB exemption you might consider using other methods such as using a phantom to show the functioning of the device. Additionally, N of 1 would be helpful is describing how the device works but would not be sufficient in showing the system accuracy.
Section 3.1 experimental setup should be moved to Methods Section.
2 in I2C is a superscript. I2C appears as I2C in a few places within the manuscript.
In line#310 why 12 minutes is needed to make a diagnosis of high temperature?
Line# 373 what do the authors mean by “No strange point is detected in the 373 measurement of Table 2.”
Author Response

(The authors gave the same response as above.)

Reviewer 3 Report
The paper presents a wearable device for real-time measurement of body temperature, environment conditions (temperature and humidity) and physical activity in simultaneous way. Two different sensors are used for body temperature. The device is also equipped with Bluetooth and Wi-Fi connectivity for data uploading to a commercial Cloud platform. The primary aim of the device is the monitoring of people subjected to quarantine restrictions due to Covid-19 pandemic. The test was done involving a single user.
The reviewer feeling is that the argument fits the scope of the journal, but the paper has too many weaknesses to be considered for publication in this version. Additional information and corrections are needed to complete the work.
In the following, some suggestions to improve the paper.
About the design, there is no evidence or demonstration that the environment sensor placed on the wearable device is not influenced by the heat produced by the user skin. A specific test to discard this situation is suggested.
It is not clear how all parameters and thresholds used in sensing data processing (e.g. Thigh=12 mins, Ts=30sec, Delta Zth = 90 mg, etc.) have been selected.
There is no comparison between the measured temperature values and the values obtained by a golden-reference device. A comparison in that respect is suggested to asses the validity of the data measured by the presented device.
The device has been tested only with one subject, for a total of 14 days. This represents a strong limit of the presented work, because the collected results does not allow to asses the real performance of the system nor to asses if the processing parameters and the method are adequately general to fit the requirements of different people to be monitored or the system has been tailored to the tested subject.
No discussion about the usability reported by the user is reported.
In many points the paper is too general, and important information for the reader are missing. For example – Line 135, How the proposed approach could improve diagnostic performance? – The ambient conditions are checked frequently (line 152). How frequently? - What “good quality” of the MCP9808 sensor means (line 190)? – Which are the LP filter characteristic frequencies (line 205) ? – How the normal body temperature threshold is automatically adjusted? - How Zacc acquisition works? Is it a 30s windowed algo?
Additionally, some sentences are unclear or contradictory. – Line 75 states the relative humidity best range is 20%-60% while at line 338 a different range is reported. – Line 124/125 seems to indicate the digital thermometer as primary sensor while the IR as secondary data source, but in the following line 172 the IR is presented as the core of the system to monitor the temperature.
No information on battery (e.g. type, capacity, etc.) have given, and considering the declared duration (45 mins) some considerations about the device power budget need to be done. Moreover, this frequent need of recharge affects dramatically the usability and the effectiveness of the device in Covid-19 monitoring.
References are adequate, but the order of citations is not correct.
An extensive revision of English language, the overall correction of typos and the uniformity of terms in the paper (e.g. Tth, TTH, ecc. ) are required.
Author Response

(The authors gave the same response as above.)

Round 2
Reviewer 1 Report
The work has been improved. Both introduction and conclusions provide an overview of the work and highlight its motivation.
The flow chart for the algorithm fits well with the text.
Author Response
Thank you for your kind comments. We attach our response file here.

Reviewer 3 Report
After the first round of revision, the quality of the paper has been improved. Most of the comments received have been addressed successfully.
For this reviewer, two points still remain open:
- How to ensure that environment readings of temperature provided by the device are not influenced by the heat produced by the user skin, considering the close position of the sensor to the user wrist. A comparative test of data collected by the on-board sensor vs an external independent environment sensor should exclude this doubt.
The new Section 2.2.3 is about the contact digital thermometer calibration, but it seems not answering to the previous question. - The minimal testing phase with only one subject. As reported in the reply letter, the authors are planning a larger test in future. It should be interested to read in the paper some information about this future work: number of subjects they intent to involve, aims of the test (e.g. to confirm the result; to test the replicability of the method on different subjects, …), etc.
Minor: line 146, repeated words typo.
The reviewer confirms that the argument fits the scope of the journal, and the work will be eligible for publication after these points will be addressed.
Author Response

(The authors gave the same response as above.)
